# Cryo EM Analysis Reveals Inherent Flexibility of Authentic Murine Papillomavirus Capsids

**DOI:** 10.3390/v13102023

**Published:** 2021-10-07

**Authors:** Samantha R. Hartmann, Daniel J. Goetschius, Jiafen Hu, Joshua J. Graff, Carol M. Bator, Neil D. Christensen, Susan L. Hafenstein

**Affiliations:** 1Department of Biochemistry and Molecular Biology, The Pennsylvania State University, University Park, PA 16802, USA; sxh739@psu.edu (S.R.H.); djg38@psu.edu (D.J.G.); jjg5869@psu.edu (J.J.G.); 2Department of Medicine, The Pennsylvania State University College of Medicine, Hershey, PA 17033, USA; 3Department of Pathology, The Pennsylvania State University College of Medicine, Hershey, PA 17033, USA; fjh4@psu.edu (J.H.); ndc1@psu.edu (N.D.C.); 4The Jake Gittlen Laboratories for Cancer Research, Hershey, PA 17033, USA; 5Huck Institutes of the Life Sciences, The Pennsylvania State University, University Park, PA 16802, USA; czb221@psu.edu; 6Department of Microbiology and Immunology, The Pennsylvania State University College of Medicine, Hershey, PA 17033, USA

**Keywords:** mouse papillomavirus, cryo EM, HPV16

## Abstract

Human papillomavirus (HPV) is a significant health burden and leading cause of virus-induced cancers. However, studies have been hampered due to restricted tropism that makes production and purification of high titer virus problematic. This issue has been overcome by developing alternative HPV production methods such as virus-like particles (VLPs), which are devoid of a native viral genome. Structural studies have been limited in resolution due to the heterogeneity, fragility, and stability of the VLP capsids. The mouse papillomavirus (MmuPV1) presented here has provided the opportunity to study a native papillomavirus in the context of a common laboratory animal. Using cryo EM to solve the structure of MmuPV1, we achieved 3.3 Å resolution with a local symmetry refinement method that defined smaller, symmetry related subparticles. The resulting high-resolution structure allowed us to build the MmuPV1 asymmetric unit for the first time and identify putative L2 density. We also used our program ISECC to quantify capsid flexibility, which revealed that capsomers move as rigid bodies connected by flexible linkers. The MmuPV1 flexibility was comparable to that of a HPV VLP previously characterized. The resulting MmuPV1 structure is a promising step forward in the study of papillomavirus and will provide a framework for continuing biochemical, genetic, and biophysical research for papillomaviruses.

## 1. Introduction

In 2011, a mouse papillomavirus was identified in a colony of nude (NMRI-Foxn1^nu^/Foxn1^nu^) mice in India and later named as Mus musculus papillomavirus 1 (MmuPV1) [1]. Although there are other identified rodent papillomaviruses, this was the first rodent papillomavirus that was found to infect laboratory strains of mice [2]. MmuPV1 is the ideal candidate for the study of native papillomavirus in the host of a common laboratory animal with abundant reagents [3].

Papillomavirus is a T = 7d icosahedral, double stranded DNA virus with a genome of ~8 kb. The capsid is composed of 72 capsomers, each containing five copies of L1, the major capsid protein. Each capsid also contains an undetermined number of L2, the minor capsid protein. There are 12 pentavalent capsomers per virus particle that have true 5-fold symmetry since they are surrounded by 5 neighboring capsomers. There are 60 hexavalent capsomers per virus particle that have pseudo 6-fold symmetry, indicating that each is surrounded by 6 neighboring capsomers that are not evenly spaced. For each copy of L1 a loop of the C-terminus forms a connecting arm by extending to a neighboring capsomer, forming a stabilizing disulfide bond, and returning to the donating capsomer. The secondary structure of L1 proteins take the form of the ubiquitous β-jellyroll made up of eight anti-parallel β-strands (BIDG and CHEF) with loop regions that extend outward from the surface of the capsid and constitute the majority of the L1 hypervariable regions [4,5].

Human papillomavirus (HPV) is epitheliotropic and its replication is tightly associated with the terminal differentiation of basal cells into keratinocytes. The restricted tropism makes production and purification of high titer virus preparations for research problematic. Although there is a tissue culture raft system to grow native HPV16 from differentiating cells, this system has not been used successfully to produce enough purified virus for structural studies [5]. Thus, researchers depend on three different virus like particles (VLP) as surrogates to study HPV in a laboratory setting: quasivirus (QV), pseudovirus (PsV), and L1-only virus like particle (L1-only VLP) [5]. All of these VLPs are expressed by plasmids that have been codon optimized for maximum production in tissue culture. QV and PsV capsids are comprised of HPV L1 and L2. QV has a packaged cottontail rabbit papillomavirus genome. PsV has one of two reporter plasmids packaged in the capsid; 5 kb-pYSEAP (secreted alkaline phosphate plasmid), or 8 kb-pCLucf (luciferase expression plasmid). L1-only VLP are made solely of the major capsid protein with no minor capsid protein or genomic material [5].

The structures of sub-viral particles have been solved to high resolution by X-ray crystallography, including the HPV16 L1 pentamer and a T = 1 capsid [6,7]. The complete T = 7d icosahedral papillomavirus capsid has previously been visualized to modest resolution by cryogenic electron microscopy (cryo EM) with icosahedral symmetry imposed during the reconstruction [4,8,9,10,11,12,13,14]. No structural studies of papillomavirus capsids with 60-fold icosahedral symmetry imposed have exceeded 4.1–4.4 Å resolution [13,14,15,16]. The structure of HPV16 QV published by Guan et al. at 4.3 Å showed there is likely inherent flexibility of the papillomavirus capsid as evidenced by the variability in capsid resolution, and differing capsid diameters [13]. In Goetschius & Hartmann et al. icosahedral subparticle extraction and correlated classification (ISECC) was developed and used for the first time on HPV16 QV that allowed for high resolution of 3.1 Å to be achieved [17]. ISECC was inspired by local reconstruction and block-based reconstruction programs which have resulted in improved resolution of other large icosahedral viruses [18,19]. ISECC provides efficient use of computational resources that allows for full pixel size processing. In addition, subparticle location metadata retains the original coordinates and location of each subparticle in the context of the icosahedral capsid.

Here, we purified a native mouse papillomavirus (MmuPV1), for which we have used ISECC to obtain high resolution maps and made a comparison for the first time between native papillomavirus and lab generated capsids. The reconstruction methods resulted in a final recombined 3.3 Å map of MmuPV1. Our approach compensated for global flexibility resulting in improved amino acid side chain density, revealed the disulfide bonds that connect the capsomers, and resolved differing capsomer conformations. Furthermore, for the first time, we detailed the interconnection between capsomers that explains why each connecting arm has a different conformation resulting in variations in the local resolution. Finally, we presented more information on the papillomavirus capsid flexibility than previous experiments revealed and established the differences between a native papillomavirus and a lab generated one.

## 2. Methods

### 2.1. Animals and Viral Infections

All mouse work was approved by the Institutional Animal Care and Use Committee of Pennsylvania State University’s College of Medicine (COM) and all methods were performed in accordance with guidelines and regulations. Hsd: NU (Outbred, Foxn1^nu/nu^) athymic mice (6–8 weeks) were obtained from ENVIGO. All mice were housed (2–3 mice/cage) in autoclaved cages within sterile filter hoods and were fed sterilized food and water in the PSUCOM BL2 animal core facility. Mice were sedated i.p. with 0.1 mL/10 g body weight with ketamine/xylazine mixture (100 mg/10 mg in 10 mls ddH_2_O) before being pre-wounded with a scalpel blade at the muzzle and tail skin sites as described previously [3,20,21]. Twenty-four hours after wounding, the mice were again anesthetized and challenged with infectious virus (1 μL of the DNA extract contains 1.4 × 10^8^ viral genome equivalents. About 1 × 10^9^ viral DNA genome equivalents were used for each infection site) at pre-wounded muzzle and tail sites using a TB needle to superficially scrape the inoculated sites. Monitoring was conducted weekly for infection at muzzle and tail and progress was documented photographically for each animal. The muzzle and tail lesions were visible around week three post infection and continued to grow. The experiments were terminated at different time points post viral infection and tail and muzzle lesions were harvested and pooled for preparation of viral suspension (Appendix A).

### 2.2. Virus Purification

Infectious virus was isolated from lesions on the muzzles and tails of infected mice [22]. In brief, at the termination of the experiment, the lesions were scraped from muzzles and tails with a scalpel blade and homogenized in phosphate-buffered saline (1 × PBS) using a Polytron homogenizer (Brinkmann PT10-35, manufactured by Kinematica AG, Malters, Switzerland) at highest speed for three minutes while chilling in an ice bath. The homogenate was spun at 10,000 rpm and the supernatant was decanted into Eppendorf tubes for temporary storage at −20 °C. Virus homogenate was then combined with 0.32 g/mL CsCl and ultracentrifuged in a swinging bucket Beckman SW-41-Ti rotor at 16 °C, 41,000 rpm for 24 h. The lower band was collected, concentrated, and buffer exchanged using a 100 KDa cutoff spin column. The sample was then added to a carbon-coated grid and stained with 2% phosphotungstic acid and analyzed for integrity and concentration using the FEI Tecnai G2 Spirit BioTwin (https://www.huck.psu.edu/core-facilities/cryo-electron-microscopy-facility/instrumentation/tecnai-biotwin-spirit) (Appendix A).

### 2.3. Cryo-EM Data Collection

The MmuPV1 sample was assessed for purity and concentration before vitrification for cryo-EM data collection on the Penn State Titan Krios (https://www.huck.psu.edu/core-facilities/cryo-electron-microscopy-facility/instrumentation/fei-titan-krios). Three microliters of the purified virus sample was pipetted onto glow-discharged R2/1 Quantifoil grids (Quantifoil Micro Tools GmbH, Jena, Germany), blotted for 2.5 s, and plunge-frozen in liquid ethane using a Vitrobot Mark IV (Thermo Fisher, Waltham, MA, USA). Vitrified grids were imaged using a Titan Krios G3 (Thermo Fisher, Waltham, MA, USA) under automated control of the EPU software. An atlas image was taken at 165× magnification, and suitable areas were selected for imaging on the Falcon 3EC direct electron detector. The microscope was operated at 300 kV with a 70 μm condenser aperture and a 100 μm objective aperture. Magnification was set at 59,000× yielding a calibrated pixel size of 1.1 Å. Four, nonoverlapping exposures were acquired per each 2-um-diameter hole of the grid with the beam in parallel mode, for an overall collection of 2859 micrographs. The total dose per exposure was set to 45 e^−^/Å^2^ (Appendix A).

### 2.4. Icosahedral Refinement

Binned icosahedral refinement was done in RELION [23]. The virus particles were picked using auto-picking with 2D templates. The particles were extracted at a 2× binned pixel size of 2.2 Å. After 2D and 3D classification 10,181 particles were used for 3D refinement. The post processed 3D refinement led to a 4.85 Å-resolution structure. Full pixel size icosahedral refinement was performed in cryoSPARC with on the fly down sampling to 1.25× [24]. The micrographs underwent full frame motion correction and CTF estimation (CTFFIND4) [25]. Micrographs were curated and sorted to reject micrographs with crystalline ice. Particles were picked using 2D templates from 348 particles. Local motion correction was performed on the particle stack and the CTF estimated micrographs. The particles underwent 2D and 3D classification in order to select only for particles containing internal (genome) density. The resulting particles then went into a homogenous refinement. The final resolution was determined by gold standard FSC threshold of 0.143.

### 2.5. Icosahedral Subparticle Extraction and Correlative Classification

Icosahedral Subparticle Extraction and Correlative Classification (ISECC) [17] is a Python-based subparticle extraction package inspired by localized reconstruction [18] and block-based reconstruction [19]. A metadata file containing icosahedrally refined particle origins and orientations is required as input. The beta version of ISECC was used with cryoSPARC metadata files [17,26]. ISECC_subparticle_extract was used after icosahedral refinement to divide each particle image into subparticles [17]. Both hexavalent and pentavalent capsomer subparticle were separately created. An initial model was made for each subparticle type using relion_reconstruct with 10,000 subparticles [23]. The subparticles were then processed in RELION v3.1 [23].

Pentavalent and hexavalent capsomers were extracted in ISECC using the following commands, respectively:
[–vector 0 147 238–roi fivefold–supersym I1–subpart_box 400]and,[–vector 85 68 245–roi fullexpand–supersym I1–subpart_box 300]

### 2.6. Local Subparticle Refinement

Pentavalent subparticles were locally refined with C5 symmetry whereas hexavalent C1 was used to refine the pentavalent subparticles. Each subparticle dataset was refined with a spherical mask of 200 Å applied to focus on the capsomer and adjoining arms. The final resolutions were determined by gold standard FSC with a threshold of 0.143 in RELION post-process. Local resolution maps were generated using RELIONs own software.

Capsomers were locally refined in RELION 3.1 beta using the following commands:
[‘which relion_refine_mpi‘–o Refine3D/job027/run–auto_refine–split_random_halves–i fivefoldsubparticles_subpart_PRIOR_invert.star–ref fivefold_initmodel_invert.mrc–firstiter_cc–ini_high 20–dont_combine_weights_via_disc–no_parallel_disc_io–preread_images–pool 100–pad 2–ctf–ctf_corrected_ref–particle_diameter 220–flatten_solvent–zero_mask–oversampling 1–healpix_order 5–auto_local_healpix_order 5–offset_range 3–offset_step 2–sym C5–low_resol_join_halves 40–norm–scale–j 1–gpu –dont_check_norm–sigma_ang 1.5]and,[‘which relion_refine_mpi‘–o Refine3D/job052/run–auto_refine–split_random_halves–i fullexpand_subpart_PRIOR.star–ref fullexpand_initialmodel_c1.mrc–firstiter_cc–ini_high 20–dont_combine_weights_via_disc–no_parallel_disc_io–preread_images–pool 100–pad 2–ctf–ctf_corrected_ref–particle_diameter 220–flatten_solvent–zero_mask–oversampling 1–healpix_order 5–auto_local_healpix_order 5–offset_range 3–offset_step 2–sym C1–low_resol_join_halves 40–norm–scale–j 1–gpu–dont_check_norm --sigma_ang 1.5]

### 2.7. Icosahedral Recombined Map

Postprocessed subparticle maps were recombined into a complete capsid using ISECC_recombine [17]. This procedure is similar to the recombination process for subparticles in Block Based Reconstruction and LocalRec [18,19,27]. Briefly, ISECC_recombine loads the subparticle maps into a numpy array and both shifts and rotates the maps to their locations in an idealized icosahedron, using regular grid interpolation in real space. This interpolation scheme allows merging of the subparticle-refined models into a single asymmetric unit.

### 2.8. Model Building

The L1 protein structure from HPV16 asymmetric unit (PDB: 7KZF) was used as in SWISS-MODEL a homology model along with the sequence of MmuPV1 L1 to make an initial protein model of MmuPV1 L1 [8,17,28]. The sequence alignment for L1 of each was performed using T-Coffee (Appendix A) [29,30]. This model of the asymmetric unit was opened in chimera and a pdb file was written out to compose of only the 5 chains of the asymmetric unit that make up the hexavalent capsomer. An additional PDB file was created that contained 5 copies of the pentavalent capsomer chain that were pieced together using 5 copies of the asymmetric unit. These PDBs were fit independently into the electron density maps in Chimera [31]. The protein structure of L1 in the hexavalent and pentavalent states were then refined in real space against the corresponding electron density maps in Phenix with geometry and secondary structural restraints [32]. The structure was visually inspected and manually refined in Coot and validated using MolProbity [33,34] (Appendix A).

### 2.9. Correlation of Locally Refined Capsomers Coordinates

Correlational analysis was performed using the ISECC_local_motions script. Briefly, this script parses the locally refined pentavalent and hexavalent capsomer star files to evaluate local deltas for subparticle origins and poses as compared to their idealized, icosahedrally derived, starting values. Coupled with the new metadata identifiers, rlnCustomVertexGroup and rlnCustomOriginXYZAngstWrtParticleCenter, this identifies deviation of each capsomer from idealized icosahedral symmetry on a per-particle basis.

Deviation in particle diameter was calculated for all particles that satisfied selection criteria, namely, a pair of pentavalent capsomers within +/− 5% (12.9 Å) of the central plane (Z = 0), where z_max_ corresponded to the particle radius as defined by the distance between the center of capsomer and the particle center. Despite a 5% tolerance being larger than necessary for a dataset of this size, it was chosen to allow for easy comparison to future datasets with a more modest particle number. This geometry minimizes contribution to distance along the Z-axis, which unlike X or Y cannot be locally refined. These capsomers are easy identified using the Z parameter within rlnCustomOriginXYZAngstWrtParticleCenter. The locally refined XY distance between polar opposite pentavalent capsomers was calculated for the 50,224 qualifying particles and compared to the Z-flattened icosahedrally derived distance, producing a difference ratio. It is important to note that this analysis cannot capture the Z-component of any diameter deviation, necessitating the selection criteria described above.

Capsomer distance analysis was conducted on all particles with a hexavalent subparticle center within 25 Å x,y distance of the whole particle center. Capsomers with a potential doppelganger within 15 Å x,y distance on the opposite face of the capsid were excluded from this analysis to avoid the risk of subparticle identity swap during local refinement. Neighboring subparticles were then identified using the rlnCustomVertexGroup parameter, allowing the geometric relationships between capsomers to be distinguished (see the four relationships listed in Figure 6b). The locally refined x,y distance for each pair of qualifying capsomers (A:F 38,238, B:Null 77,566, C:D 75,468, E:E 42,228) was divided by the icosahedrally derived x,y distance (dropping the z coordinate) to produce an in-plane ratio corresponding to contraction or expansion along the given axis.

All statistical analysis was completed with the use of JMP Pro statistical software version 15.0.0 (SAS Institute Inc., Cary, NC, USA) and evaluations between virus types was completed using unequal variances.

## 3. Results

### 3.1. Native Papillomavirus Icosahedral Refinement Stalled at 4.4 Å Resolution

Cryo EM micrographs showed virus particles ~55 nm in diameter with discernable capsomers subunits comprising the capsids. There was a mixed population of packaged native virus capsids along with those that were devoid of genomic density. Consistent with previous studies, there was variation in particle size as well an occasional rod like structure (Appendix A). A standard reconstruction pipeline classified 11,700 particles into a majority population of capsids that contained internal density consistent with a packaged genome (10,181) and a minority (1519) of particles that had no internal density present and were excluded from further processing (Figure 1A,B). Virus particles containing internal density were selected for a reconstruction with icosahedral symmetry imposed, which resulted in a 4.4 Å resolution map. Within this icosahedral map α-helices and a minimal amount of β-strand separation was distinguished. However, at this resolution the side chains cannot be visualized, and the structure cannot be reliably built.

### 3.2. There Are Resolution Differences between Hexavalent and Pentavalent Capsomers

Local resolution of the icosahedrally refined virus capsid showed capsomers in the hexavalent environment had better resolution than the pentavalent capsomers. The central density of all capsomers, comprised of β-strands, was uniform between the two capsomer conformations conferred by the hexavalent and pentavalent environments. This core β-strand density had the highest resolution of the capsid map (Figure 1C–E). The lowest resolution areas of the capsid corresponded to the surface loops and the C-terminal extensions that connect the capsomers. Thus, the capsomer β-jellyroll secondary structure is relatively stable and it is likely the flexibility in the connecting arm and loop regions that causes the overall modest map resolution.

### 3.3. High Resolution Capsomers

To overcome the inherent flexibility of the icosahedral virus and improve the resolution, subparticles were designated based on the capsomeric architecture distinct to papilloma and polyomaviruses. Thus, twelve pentavalent subparticles and sixty hexavalent subparticles were extracted from each original capsid using our program Icosahedral Subparticle Extraction & Correlated Classification (ISECC) (Figure 2A,B) [17]. ISECC extracts subparticles while retaining the original location for each as retraceable metadata.

After extraction the pentavalent and hexavalent subparticles were refined independently imposing five-fold symmetry averaging (C5) and no symmetry averaging (C1), respectively. Refinement of the pentavalent and hexavalent capsomers resulted in subparticle maps with local resolution ranging from 3.2 Å to 3.6 Å (Figure 2C,D). The subparticle refinements were recombined into an icosahedral capsid with 3.3 Å resolution overall (ISECC_Recombine), a dramatic improvement over the original 4.4 Å map (Figure 3A,B). The high resolution of the recombined map allowed for the full asymmetric unit of MmuPV1 to be built for the first time.

### 3.4. Hexavalent Capsomers Are Asymmetric

The hexavalent capsomers, reconstructed with C1 symmetry, attained higher resolution than the pentavalent capsomers, that had C5 symmetry imposed. In the hexavalent capsomer the local resolution maps exhibited obvious asymmetry. The highest resolution mapped to the threefold that is comprised of repeating C -> D and D -> C arm connections, suggesting that this is the most stable area of the virus (Figure 2). High resolution areas were seen reaching away from the fivefold, which is surrounded by single arm B -> F connections. The twofold resolution fell in between that of the fivefold and threefold, even though the twofold is the only position where there is same conformation of donating and receiving arms (E -> E and E -> E) (Figure 2C).

### 3.5. MmuPV1 Major Capsid Protein Visualized for the First Time

The HPV16 L1 structure (PDB ID 7KZF) was mutated to the equivalent MmuPV1 L1 primary sequence and used to initiate the build into the recombined, refined, and validated map (Appendix A) [13,28]. The L1 protein structure was reliably visualized from Thr16–Ala481 (Figure 3C,D). The build of the L1 protein revealed the typical papillomavirus secondary structure elements of BIDG and CHEF anti-parallel beta strands making up the L1 core within each capsomer. All of the loop regions at the surface of the virus capsomers were also resolved. Despite the sequence deviation between HPV and MmuPV1 (53% identity) there is structural conservation of the secondary, tertiary, and quaternary structures.

Each copy of L1 has a C-terminal loop region spanning Trp 406–Trp 451 and five of these connecting arms extend from each capsomer, independent of whether the capsomer is in the hexavalent or pentavalent icosahedral environment. Each of these loop regions extends to interact with a neighboring capsomer and these interactions are stabilized by a disulfide bond between Cys 432 of the connecting arm and Cys 173 of the neighboring capsomer (Figure 4) [20,21,35]. Despite each copy of L1 being chemically identical, there are six unique disulfide bonds that occur, one per L1 chain comprising the asymmetric unit. The resolution of the map density for the connecting arms ranged from 3.3–3.8 Å with some spots of weak density suggesting that the connecting arms are flexible.

### 3.6. Putative L2 Density Is More Prevalent in Pentavalent Environments

After refining L1, there was notable unfilled density on the interior of the capsomers. The density was strand like with protruding knobs, consistent with a protein of approximately 6 amino acid residues each. The density (6 aa) was found flanking an L1 loop region in the capsomer core (Figure 5). This putative L2 density was present underneath pentavalent and hexavalent capsomers, but the magnitude of the density was stronger within the pentavalent capsomer.

This probable L2 density is almost identical to what has been observed in the codon-optimized, lab generated, HPV16 quasivirus [17]. As in Goetschius & Hartmann et al., 2021 we observed the same capsomer environment preference for the putative L2 density. Finding the similar non-L1 densities with the same pentavalent-favored type of incorporation into a native virus capsid further supports the identity of this non-L1 density as L2. The fragmented nature of these non-L1 densities also suggested that L2 may have long disordered stretches between the icosahedrally ordered regions that were resolved in the high-resolution maps.

### 3.7. Capsids Have Imperfect Icosahedral Symmetry

In addition to the improved resolution, local subparticle refinement allowed a determination of the true capsomer centers and orientations. The program ISECC extracts subparticles from a map that has been generated by imposing perfect icosahedral symmetry. However, after extraction each subparticle is refined individually to find the actual capsomer center and position. ISECC retains both the actual location and the icosahedrally enforced location for each subparticle as retraceable metadata. A comparison of the actual capsomer centers relative to each icosahedrally idealized parameter illustrates capsomer movement away from true icosahedral symmetry. This movement suggests flexibility. The new locations of the accurate capsomer centers allowed us to evaluate capsid flexibility in two different ways using established methodology [17]: (1) inter-capsomer movement, defined as the quasi-independent movement between capsomers, and (2) the range of whole particle diameters, defined as the distance between two polar opposite pentavalent capsomers.

Inter-capsomer movement was measured for all qualifying particles with a hexavalent capsomer within 25 Å of the x,y distance from the particle center (see Methods). This parameter allowed measurement of lateral, in-plane movement of capsomers and limited out of plane motion (along the z axis) due to defocus that cannot be recovered. From each qualified hexavalent capsomer, the distances to all six neighboring subparticles were calculated based on their locally refined (non-icosahedrally forced) coordinates. The locally refined distances between capsomers varied from the idealized (icosahedrally forced) values by a standard deviation of approximately 2.5%. Each of the four unique arm connections was defined by its contribution to the centered hexavalent capsomer making the four following patterns: on the fivefold, hexavalent chain A to pentavalent chain F (A:F), surrounding the fivefold, hexavalent chain B to a neighboring hexavalent capsomer with no reciprocation (B:Null), at the three fold, hexavalent chain C to hexavalent chain D (C:D), and at the twofold, hexavalent chain E to hexavalent chain E (E:E). Each type of arm connection showed a similar pattern of deviation with the ability to expand and contract by 3–4% (95th percentile range) (Figure 6A–D).

To understand global effects of capsid flexibility, whole particle diameter was measured for all qualifying particles with two polar opposite pentavalent capsomers that were within the same z-plane allowing a 5% tolerance from z = 0 [17]. A diameter ratio was calculated for each qualifying capsid based on the icosahedrally averaged map diameter of 575 Å, and the 95th percentile (0.980–1.014) corresponded to capsid diameters ranging from 563–583 Å (Figure 6E).

### 3.8. MmuPV1, a Native Papillomavirus Is Globally Flexible, but Less So Than HPV16, Quasivirus

Due to the amount of flexibility observed in MmuPV1, a comparison was made between the native virus and the codon-optimized, lab generated, HPV16 quasivirus that had been previously published [17]. Using the same methods and enforcing the same parameters, the two datasets were compared. It is important to note that due to the different particle number of each of these datasets, to perform a valid comparison we used statistical tests that evaluated unequal variance. We compared the datasets across five parameters: the four possible intercapsomer connections (A:F, B:Null, C:D, and E:E), and overall diameter (Figure 6, Appendix A).

For both viruses independently, the four types of intercapsomere connections were remarkably similar despite the differences in the connections. Each of the MmuPV1 intercapsomere connections was closer together, showing less variance than the HPV16 quasivirus data. Each virus type had a mean close to 1, but the spread of the data was different. Within the 95th percentile, the intercapsomer connections of HPV16 quasivirus had the ability to expand and contract by 6–7% (about 7–9 Å) whereas MmuPV1 only had the ability to expand or contract by 3–4% (about 3–5 Å).

For overall diameter differences we had an interesting finding that the data for MmuPV1 had more variance than that of HPV16 quasivirus, meaning that MmuPV1 capsids had a wider range of diameter distances. Within the 95th percentile, the overall dimeter of HPV16 quasivirus and MmuPV1 had the ability to expand or contract by about 8 Å and 10 Å, respectively. It was also noted that the mean for MmuPV1 was shifted below 1 indicating there is a higher possibility for the virus to appear more contracted with a smaller diameter size than there was for expansion.

## 4. Discussion

Previously, reconstructions of papillomaviruses with full icosahedral symmetry imposed have produced maps limited to modest resolution; however, recently subparticle reconstruction approaches have been successfully used to obtain high resolution [16,17,36]. Consistent with this finding, we found that for the native papillomavirus the icosahedrally enforced reconstruction did not advance beyond 4.4 Å resolution. Using a subparticle reconstruction approach to refine each capsomer individually allowed us subsequently to build the subparticles into a recombined virus capsid map with 3.3 Å resolution. The high resolution of the map allowed us to visualize the L1 asymmetric unit of MmuPV1 for the first time.

The high-resolution map and new build of L1 allowed us to identify protein-like density that was unfilled by L1, which likely corresponds to the minor capsid protein L2. The clear strand separation between the continuous L1 density and the ordered fragments of putative L2 suggests that the fragments of L2 can be seen because they are interacting or stabilized by the flanking regions of L1. It was also noted that the L2 density was stronger in the pentavalent environment than hexavalent, but the L2 density was never as strong as the L1 density. The magnitude of this putative L2 density suggests that any of the potential sites could be occupied. However, due to the location and the known size of L2, steric collisions would allow only one or two copies of L2 to fit beneath each capsomer.

Remarkably, the regions of L2 that were found in the native MmuPV1 capsids were also seen in the codon-optimized, lab generated, HPV16 quasivirus sample [17]. The consistency between these two findings further supports that the protein strand densities unfilled by L1 are in fact L2. The variability in previous L2 stoichiometry studies supports the model that L2 is asymmetrically incorporated into the capsids, and that the overall number of L2 per capsid may vary [9,37,38,39]. Our studies suggest that for both MmuPV1 and HPV16 quasivirus, L2 is preferentially incorporated into the pentavalent capsomers. Furthermore, upon incorporation into the capsid, L2 maintains significant disordered stretches and the only resolvable L2 protein regions are those interacting with and being stabilized by L1.

Upon gross examination in negative stain or cryo EM micrographs, diameter differences of papillomavirus capsids are obvious. Earlier attempts were made to classify and sort HPV particles by diameter [13]. Although moderately successful as indicated by the resulting 4.3 Å map, it seems likely that the capsid diameters are not going to fall into discrete individual classes, but rather exist along a continuum describing a range of diameters. More recently, ISECC_local_motions has been used to quantify the magnitude of flexibility of the particles according to particle diameter, and also to evaluate the movement between capsomers. The flexibility analysis performed here provided the opportunity to compare a native papillomavirus, MmuPV1, that was extracted and purified directly from infected mice, to a codon-optimized, HPV16 quasivirus, that is lab generated HPV16 capsid proteins with a cottontail rabbit papillomavirus genome. Nevertheless, the authentic virus does present several advantages serving to open up other experiments in the Papillomavirus field. As an authentic, infectious particle it can be employed in studies of entry and uncoating. It may prove a valuable tool for use in live cell imaging to track early stages of infection.

Although the diameter differences between native MmuPV1 and the lab generated HPV16 were negligible, the intercapsomere distances were larger for quasivirus than the native virus, indicating that the native virus was less flexible, i.e., more stable. This finding suggests that all papillomavirus capsids have inherent flexibility and that the flexibility that was seen in HPV16 quasivirus could not solely be attributed to unauthentic genome or codon-optimized structural proteins.

Single particle cryo EM reconstructions take the average of each particle within a vitrified sample, although each of these particles may represent a state within an equilibrium of particles in a solution. These, states provide a source of experimental capsid dynamic data that could contribute to whole-capsid dynamics. The work presented here supports the model previously proposed that papillomavirus capsids are dynamic and in a state of constant flexing. This model is supported by capsomer core stability from a beta-jellyroll motif, and flexibility that stems from long loop regions responsible for connecting capsomers. The intercapsomer flexing is global and variable and does not have coordinated movements. The movement between two capsomers is quasi-independent as they are part of the global network of 72 capsomers tethered together by disulfide bonds. It is likely that for particles even of the same diameter there are differences and different points of the virus flexing to create imperfect icosahedral symmetry. Overcoming these variations is why the subparticle reconstruction focused on capsomers to allow for high resolution to be obtained by centering the most stable element for refinement.

We can now say that papillomaviruses are flexible despite being native or lab generated. The ability for these particles to flex suggests a multitude of benefits for virus survival. As a virus that thrives in mucosal tissue, the capsid flexibility could be a way to protect the virus from the environment and control the hydration of the capsid as well as protect against pH and temperature changes [1,3,40,41,42,43]. It may also be likely that mechanistically capsid movements can aid in the conformational changes that occur during entry [44,45,46]. The ability to expand and contract may contribute to the tight packing of virus capsids seen within the nucleus during replication [47]. Finally, flexing could be a way to alter antigenic sites and inhibit antibody recognition. Overall, capsid flexibility is a dynamic factor and necessary contribution to papillomavirus function.

## Figures and Tables

**Figure 1 viruses-13-02023-f001:**
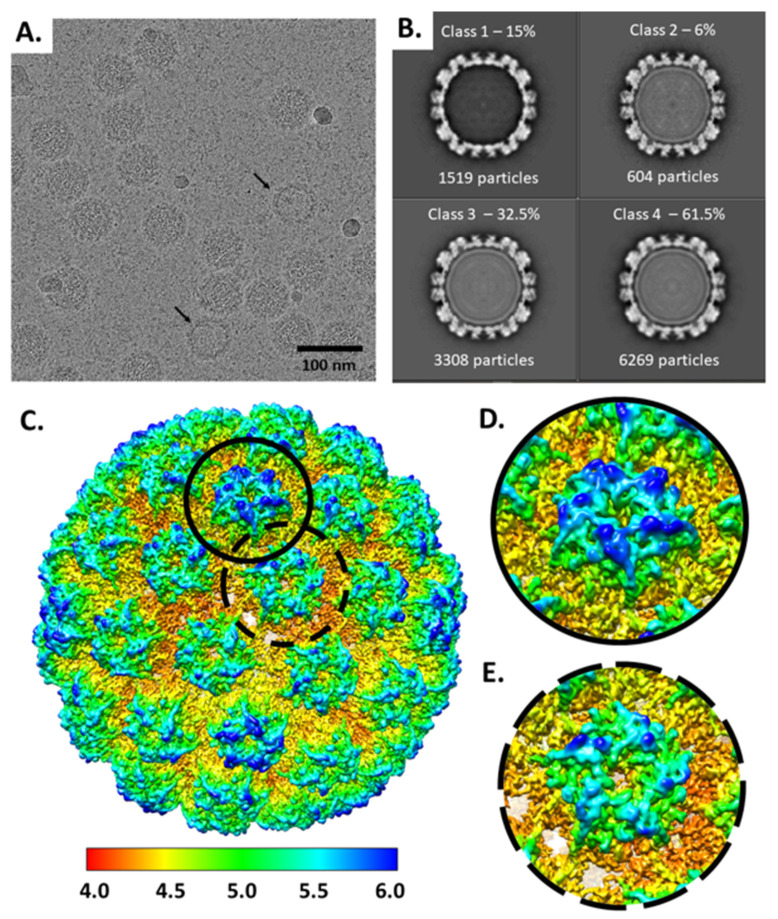
Cryo EM Icosahedral Reconstruction. (**A**) Representative micrograph 386/2,859 at 14,475 Å defocus with a slight grainy background possibly due to residue from the purification gradient (Methods). Some particles are devoid of genome (black arrows). (**B**) Three-dimensional classification sorted into 4 classes. Class 1 accounts for 15% of the particles and has a bright white capsid density shell, with a dark core corresponding to lack of internal density. Due to the low particle number, a separate reconstruction of empty capsids was not pursued in this study. Classes 2–4 that have gray internal density denoting a packaged genome were selected and used for all further processing. (**C**) Local resolution of the icosahedral reconstruction displayed surface rendered. Red represents highest resolution values (4.0 Å) and dark blue represents lowest resolution values (6.0 Å). The pentavalent capsomer (solid black line) and the hexavalent capsomer (dotted black line) are shown as zoomed views (**D**,**E**).

**Figure 2 viruses-13-02023-f002:**
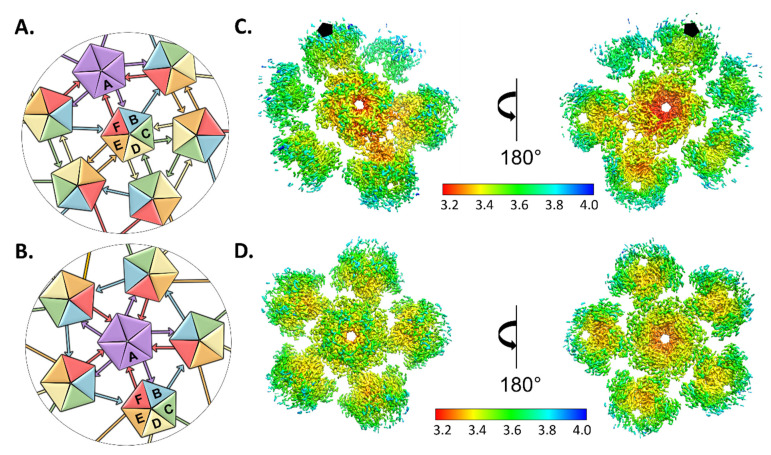
Local refinement of hexavalent and pentavalent capsomers. (**A**,**B**) With this analysis the cartoon has been modified from Goetschius and Hartmann et al., 2021 to display accurately the donating and receiving connecting arms [17]. Illustrated graphically by cartoons of pentavalent and hexavalent capsomers, the papillomavirus capsid asymmetric unit is made up of six L1 chains, labeled chain A–F and colored purple, blue, green, yellow, orange, and red, respectively. (**A**) Each A chain of the pentavalent capsomer makes a connection with chain F of the neighboring hexavalent capsomer, showing fivefold icosahedral symmetry. The pentavalent capsomer is surrounded by single arm connections of chain B to chain F in a counterclockwise ring contribution. (**B**) The threefold is comprised of C and D chains with each C chain (green) donating to chain D and D donating to C. Each E chain (orange) contributes to a neighboring E chain, showing twofold symmetry and this is the only case where the same arm conformation connects two capsomers. (**C**,**D**) Surface rendered subvolumes colored according to local resolution (color key) with exterior views (left) and views flipped 180° for interior view (right). Highest resolution (red) in both environments is seen at the interior core of the capsomer with resolution diminishing through the connecting arms and at the outer edges (**C**). Pentavalent subvolumes are centered on the fivefold icosahedral symmetry axes, whereas a black pentagon identifies the fivefold axis location in the hexavalent capsomer (**D**).

**Figure 3 viruses-13-02023-f003:**
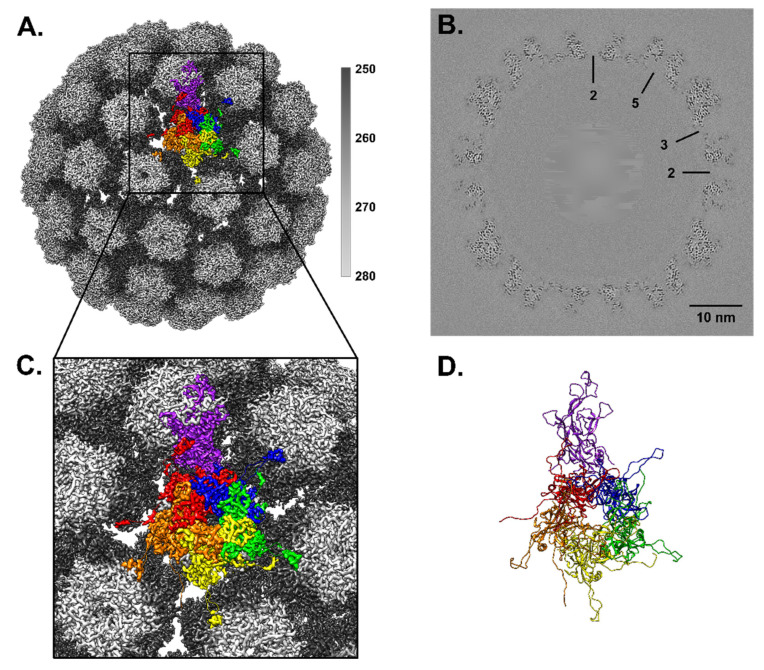
High resolution recombined map. (**A**) The high resolution recombined MmuPV1 map is surface rendered and colored radially (see key) with one copy of the asymmetric unit fit into the map and colored by chain using the same color code as Figure 2. (Chain A–F: purple, blue, green, yellow, orange, and red). (**B**) The high quality of the map is displayed in the central section with symmetry axes labeled. (**C**) Zoom view from panel A illustrates this first build of L1 proteins comprising the asymmetric unit. There is continuous density for all chains. (**D**) L1 protein ribbon rendering of the asymmetric unit.

**Figure 4 viruses-13-02023-f004:**
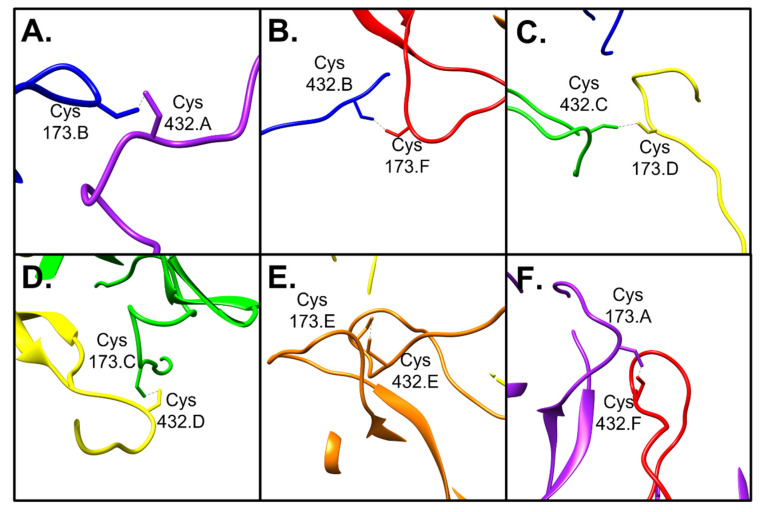
Disulfide bonds connect capsomers. Each copy of L1 has a connecting arm that extends to a neighboring capsomer where a disulfide bond is formed. (**A**–**F**) Disulfide bonds made between Cys 432 of the connecting arm, and Cys 173 of the receiving capsomer. A–F, respectively represent each connecting arm connection of chains A–F (colored as in Figure 2 and Figure 3, purple, blue, green, yellow, orange, and red, respectively).

**Figure 5 viruses-13-02023-f005:**
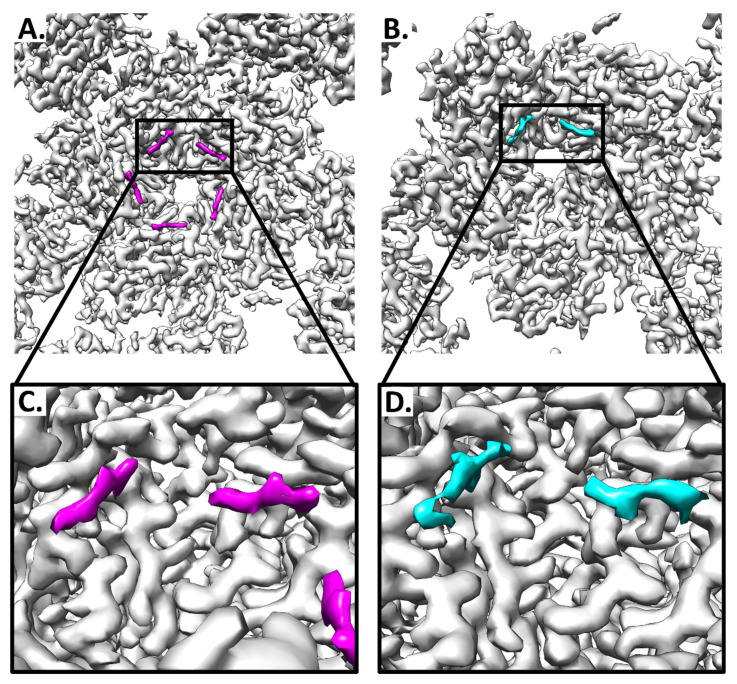
Non-L1 protein density maps to the interior of pentavalent and hexavalent capsomers. The recombined capsid map is colored (gray) within 2 Å of the continuous L1 protein chain. The discontinuous protein volumes (magenta and cyan) are unfilled by L1. (**A**) The pentavalent capsomer has 5 stretches of putative L2 density, each of which flanks one of the 5 copies of chain A that make up the pentavalent capsomer. (**B**) The hexavalent capsomer has 2 distinct stretches of putative L2 density, each flanking chain C and D of the hexavalent capsomer at the threefold. (**C**,**D**) Zoom of the putative L2 protein density from each capsomer environment.

**Figure 6 viruses-13-02023-f006:**
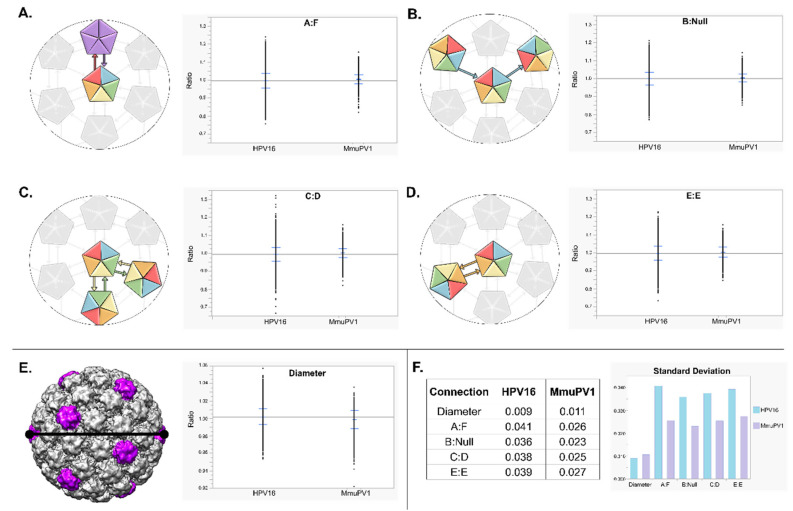
Intercapsomer and diameter distances of MmuPV1 and HPV16 QV. (**A**–**D**) Intercapsomer distance comparisons. (**A**–**D**, left) Cartoon depiction of the intercapsomere connections, A:F, B:null, C:D, and E:E, respectively. (**A**–**D**, right) Graphical representation of intercapsomere distances for all qualifying particles. For all intercapsomere distances HPV16 QV had a much wider spread of data with ratios ranging from 0.7 to 1.3, while MmuPV1 had ratios ranging from 0.8 to 1.2. (**E**, left) Cartoon depiction of the entire icosahedral virus showing the diameter is measured between two polar-opposite pentavalent capsomers (purple). (**E**, right) Graphical representation of diameter distances for all qualifying particles. HPV16 QV had a more even spread of the data with ratios ranging from 0.95 to 1.05, while MmuPV1 had ratios ranging from 0.94 to 1.04. (**F**) Standard deviation comparison between all measurement types for HPV16 QV and MmuPV1.

## Data Availability

The MmuPV1 structures of the 3.3 Å recombined map (EMDB: EMD-24741) have been deposited in the EM database (http://www.emdatabank.org/). Coordinates for the atomic model of the asymmetric unit of MmuPV1 L1 proteins (PDB: 7RYJ) have been deposited in the protein data bank (https://www.rcsb.org/). ISECC, our custom software for subparticle extraction and correlated classification, is available on GitHub (https://github.com/goetschius/isecc).

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
