# Peer review of "Cryo EM Analysis Reveals Inherent Flexibility of Authentic Murine Papillomavirus Capsids"

_viruses, 2021, doi:10.3390/v13102023_

Round 1

Reviewer 1 Report

The manuscript by Hartmann et al reports the cryo EM analysis of murine papillomavirus capsids. MmuPV1 is a valuable animal model for the study of PV and extending structure analyses to this virus therefore valuable. Due to heterogeneity and structural flexibility past resolution of HPV particles were limited. The authors utilize icosahedral subparticle extraction and correlated classification (ISECC), which was previously developed by the same authors, to achieve a 3.3Å resolution. This allowed them to build asymmetric unit and identify putative L2 density.

Overall, a well-executed and timely study. The paper is well written. It would benefit from more references to substantiate statements such as referring to the existence of disulfide bonds (line 183ff) and several statements in the discussion (previously proposed … line 347; conformational changes occurring during … (line 363) and others).

Author Response

Reviewer #1:

The manuscript by Hartmann et al reports the cryo EM analysis of murine papillomavirus capsids. MmuPV1 is a valuable animal model for the study of PV and extending structure analyses to this virus therefore valuable. Due to heterogeneity and structural flexibility past resolution of HPV particles were limited. The authors utilize icosahedral subparticle extraction and correlated classification (ISECC), which was previously developed by the same authors, to achieve a 3.3Å resolution. This allowed them to build asymmetric unit and identify putative L2 density.

Overall, a well-executed and timely study. The paper is well written. It would benefit from more references to substantiate statements such as referring to the existence of disulfide bonds (line 183ff) and several statements in the discussion (previously proposed … line 347; conformational changes occurring during … (line 363) and others).

As suggested, we have added references referring to the disulfide bonds in the results section (line 188) and we have added additional references to the discussion throughout, especially the last paragraph of the discussion (lines 360-369).

Reviewer 2 Report

The authors present a robust and innovative high-resolution structural study of a native papillomavirus isolated from infected animals. They determined a high-resolution structure by single-particle cryo-EM of the mouse papillomavirus (MmuPV1) isolated from infected mice. They used innovative subparticle averaging to obtain novel resolutions of the capsid to 3.3 Angstroms. The resulting high-resolution structure allowed the authors to build the L1 MmuPV1 asymmetric unit  for the first time and identify putative L2 density. They also characterized the flexibility or breathing of the L1 capsid which maybe important for proper capsid function. This great work will advance the broad field of structural virology and particularly the field of papillomavirus structure. There are a few minor suggestions for the paper:

Line 99 “as well an occasional rod like structure.”  Can an image of a rod-like structures be shown? Could it be another panel in figure 1 or inset for figure 1A?

Also, in Figure 1:

Could the authors comment/discuss on the grainy background in Figure 1A.

Is this nonspecific macromolecules or is it free/disassembled L1 capsomers?

Could the authors discuss how having the structure and this mouse system could open up other experiments in the papillomavirus field. For example can the capsids be used to study virus assembly, disassembly and entry/uncoating?  For the above, maybe two or three sentences in the discussion.

Also, what’s the resolution of the empty capsid in Fig. 1B? Could this be studied in the future?

Are their plans to get class I (-DNA genome ) to higher resolution or is this technically challenging? Please briefly discuss (1-2 sentences).

Line 396 to 398 “The sample was then added to a carbon-coated grid and stained with 2% phosphotungstic acid and analyzed for integrity and concentration using the FEI Tecnai G2 Spirit BioTwin”. Can a negative-stain image be put as a supplemental figure?

Line 477 479  “The L1 protein structure from HPV16 asymmetric unit (PDB: 7KZF) was used as in SWISS-MODEL a homology model along with the sequence of MmuPV1 L1 to make an initial protein model of MmuPV1 L1[8,17,20].”

It would be helpful to the broader papillomavirus filed if a supplemental figure showing the sequence alignment between HPV16 L1 and MmuPV1 L1 protein were shown. Scientists in the field could see how conserved or diverged the proteins are in terms of primary sequence.

Are Supp. Figure 1. and Supp. Figure 2. referred to in the main text/methods?

For example should Supp. Figure 2. be referenced at the end of  Line 386.

Author Response

Reviewer #2:

The authors present a robust and innovative high-resolution structural study of a native papillomavirus isolated from infected animals. They determined a high-resolution structure by single-particle cryo-EM of the mouse papillomavirus (MmuPV1) isolated from infected mice. They used innovative subparticle averaging to obtain novel resolutions of the capsid to 3.3 Angstroms. The resulting high-resolution structure allowed the authors to build the L1 MmuPV1 asymmetric unit for the first time and identify putative L2 density. They also characterized the flexibility or breathing of the L1 capsid which maybe important for proper capsid function. This great work will advance the broad field of structural virology and particularly the field of papillomavirus structure. There are a few minor suggestions for the paper:

Line 99 “as well an occasional rod like structure.”  Can an image of a rod-like structures be shown? Could it be another panel in figure 1 or inset for figure 1A?

A representative micrograph showing the occasional rod like structure has been added as supplementary figure 1.

Also, in Figure 1:

Could the authors comment/discuss on the grainy background in Figure 1A.

Is this nonspecific macromolecules or is it free/disassembled L1 capsomers?

The appearance of the background is not unusual for our images of freshly purified papillomavirus. The most likely cause of the graininess is due to cesium that was not completely removed during the final stages of purification. This possibility has been added to the figure legend to describe the background. The graininess is almost certainly not disassembled L1 capsomers as these appear more regular when we do see them in degraded samples that have been stored over time. Thus we use only freshly purified samples. We have never visualized free capsomers in freshly purified samples such as this one due to the gradient purification (any small contaminating proteins do not band with the capsids).

Could the authors discuss how having the structure and this mouse system could open up other experiments in the papillomavirus field. For example can the capsids be used to study virus assembly, disassembly and entry/uncoating?  For the above, maybe two or three sentences in the discussion.

Several suggestions were worked into the discussion to indicate advantages for having an authentic infectious virus for further studies.

Also, what’s the resolution of the empty capsid in Fig. 1B? Could this be studied in the future?

Are their plans to get class I (-DNA genome ) to higher resolution or is this technically challenging? Please briefly discuss (1-2 sentences).

The Fig. 1 B panel that shows the small subset of empty particles does not have a resolution associated with it since it is merely a classification and not a reconstruction. With such a small number of capsids, a reconstruction is not feasible. We have included more explanation (line 102 and Fig 1 legend) that due to low particle number, a reconstruction of these empty particles was not pursued and they were excluded from the reconstruction.

Line 396 to 398 “The sample was then added to a carbon-coated grid and stained with 2% phosphotungstic acid and analyzed for integrity and concentration using the FEI Tecnai G2 Spirit BioTwin”. Can a negative-stain image be put as a supplemental figure?

A representative image from the negative stain screening has been added as supplementary figure 4.

Line 477 479  “The L1 protein structure from HPV16 asymmetric unit (PDB: 7KZF) was used as in SWISS-MODEL a homology model along with the sequence of MmuPV1 L1 to make an initial protein model of MmuPV1 L1[8,17,20].”

It would be helpful to the broader papillomavirus filed if a supplemental figure showing the sequence alignment between HPV16 L1 and MmuPV1 L1 protein were shown. Scientists in the field could see how conserved or diverged the proteins are in terms of primary sequence.

The sequence alignment has been added as supplementary figure 5.

Are Supp. Figure 1. And Supp. Figure 2. Referred to in the main text/methods?

These and all additional supplemental figures have been updated and are now referred to appropriately in the text.

For example should Supp. Figure 2. be referenced at the end of Line 386.

This reference has been added.